# Identification of Crucial Genes and Regulatory Pathways in Alfalfa against Fusarium Root Rot

**DOI:** 10.3390/plants12203634

**Published:** 2023-10-21

**Authors:** Shengze Wang, Haibin Han, Bo Zhang, Le Wang, Jie Wu, Zhengqiang Chen, Kejian Lin, Jianjun Hao, Ruifang Jia, Yuanyuan Zhang

**Affiliations:** 1Key Laboratory of Biohazard Monitoring and Green Prevention and Control for Artificial Grassland, Ministry of Agriculture and Rural Affairs, Institute of Grassland Research of CAAS, Hohhot 010010, China; wangzhezhe96@163.com (S.W.);; 2School of Food and Agriculture, University of Maine, Orono, ME 04469, USA

**Keywords:** *Fusarium acuminatum*, soluble sugar, soluble protein, malondialdehyde, transcriptional factor

## Abstract

Fusarium root rot, caused by *Fusarium* spp. in alfalfa (*Medicago sativa* L.), adversely impacts alfalfa by diminishing plant quality and yield, resulting in substantial losses within the industry. The most effective strategy for controlling alfalfa Fusarium root rot is planting disease-resistant varieties. Therefore, gaining a comprehensive understanding of the mechanisms underlying alfalfa’s resistance to Fusarium root rot is imperative. In this study, we observed the infection process on alfalfa seedling roots infected by *Fusarium acuminatum* strain HM29-05, which is labeled with green fluorescent protein (GFP). Two alfalfa varieties, namely, the resistant ‘Kangsai’ and the susceptible ‘Zhongmu No. 1’, were examined to assess various physiological and biochemical activities at 0, 2, and 3 days post inoculation (dpi). Transcriptome sequencing of the inoculated resistant and susceptible alfalfa varieties were conducted, and the potential functions and signaling pathways of differentially expressed genes (DEGs) were analyzed through gene ontology (GO) classification and Kyoto Encyclopedia of Genes and Genomes (KEGG) enrichment analysis. Meanwhile, a DEG co-expression network was constructed though the weighted gene correlation network analysis (WGCNA) algorithm. Our results revealed significant alterations in soluble sugar, soluble protein, and malondialdehyde (MDA) contents in both the ‘Kangsai’ and ‘Zhongmu No. 1’ varieties following the inoculation of *F. acuminatum*. WGCNA analysis showed the involvement of various enzyme and transcription factor families related to plant growth and disease resistance, including cytochrome P450, MYB, ERF, NAC, and bZIP. These findings not only provided valuable data for further verification of gene functions but also served as a reference for the deeper explorations between plants and pathogens.

## 1. Introduction

Alfalfa (*Medicago sativa* L.) is a perennial leguminous forage grass with a long history of cultivation, wide distribution and planting, and high utilization value, which is known as the ‘king of forage grass’ [1]. China is the second largest producer in the world, accounting for 15.1% of the global alfalfa cultivation [2,3]. There has been a noteworthy shift in the dietary preferences of the Chinese populace in recent years, marked by a rising appetite for animal-based foods such as beef and mutton, leading to an increased demand for feed grains. Consequently, we are witnessing a notable expansion in alfalfa cultivation, driven by adjustments in the industrial landscape and structural reform of the supply side [4,5]. However, despite the expansion of the alfalfa planting acreage, the supply of alfalfa products continues to outpace demand.

Diseases are the main constraint in alfalfa production. They have a detrimental impact on both quality and yield, compounding other adverse effects [6]. Survey data reveals that root rot is responsible for the loss of approximately 20% of the annual alfalfa yield in the world, with some regions suffering even greater losses, reaching a 40% loss [7,8]. To effectively manage the disease, breeding of disease-resistant varieties has been emphasized [9,10,11,12,13,14,15]. The use of genetic and genomic technologies can provide valuable insights into the underlying mechanisms associated with alfalfa’s response to pathogens [16]. This contributes to a better understanding of the complexity of plant–pathogen interactions and establishes a robust scientific foundation for improving the disease resistance of alfalfa.

*Fusarium* spp. is a soil-borne pathogen that mainly overwinters in the form of mycelia and spores in diseased residues, infected seeds, and soil [17]. *Fusarium* spp. usually invades the host via roots injured by mechanical damage or environmental stresses [18]. The chlamydospores of *Fusarium* spp. are the primary inoculum and can survive in the soil for decades. When environmental conditions are favorable for the infection, the chlamydospores germinate and launch new infections within the plant host. Conidia are secondary inoculum and can survive on the surfaces of infected plants and facilitate disease spread to neighboring plants. Therefore, both chlamydospores and conidia of *Fusarium* spp. play an important role in the occurrence and spread of the disease [19,20,21]. 

Infection of plants by *Fusarium* spp. elicits a cascade of disease-resistant reactions in plants, including changes in major energy-associated substances (soluble sugar and soluble protein), changes in substances associated with oxidative stress (such as malondialdehyde, hydrogen peroxide, and flavonoids), and shifts in the activities of defense-related enzymes (such as SOD, POD, and CAT) [22,23,24,25,26]. *Fusarium* invasion can also induce the expression of disease-resistance genes. For example, *F. acuminatum* infection has been shown to induce the onset of root rot disease in *Lycium barbarum* by the up regulation specific genes and regulatory signaling pathways [27]. Additionally, the purified recombinant protein PnPR10-2 inhibits remarkable efficacy in inhibiting root rot in *Panax notoginseng* [28]. In the context of alfalfa, inoculation of *F. proiiferatum* L1 into both resistant and susceptible alfalfa clones results in distinct patterns of gene expression. This divergence becomes evident at both 24 h and 7 days post inoculation (dpi), with numerous genes implicated in PAMP-triggered immunity (PTI) and effector-triggered immunity (ETI) showing altered expression levels. Moreover, the response to infection also entails the activation of transcription factors belonging to the bHLH, SBP, AP2, WRKY, and MYB families [29].

Transcriptome sequencing (RNA-seq) is a high-throughput sequencing technique employed to evaluate the expression of RNAs within specific cells and tissues at defined time points [30]. This method not only provides insights into the expression levels of specific functional genes but also offers a comprehensive overview of overall gene expression patterns [31]. RNA-seq can be used to analyze the mechanisms underlying plant–pathogen interactions and explore the genes that play pivotal roles in unraveling the intricate mechanisms governing plant–pathogen interactions and dissecting the genes and metabolic pathways related to pathogenicity and disease resistance. 

Weighted gene co-expression network analysis (WGCNA) is an algorithm introduced by Langfelder and Horvath in 2008 [32], and it has found extensive application in the analysis of biological data. It mitigates the shortcomings of conventional differential gene screening methods, which often overlook pivotal molecules within regulatory processes and struggle to provide a holistic understanding of the biological system. WGCNA is widely used in the analysis of pathogen–plant interactions. 

As the widespread application of disease-resistant varieties is the most effective strategy to control alfalfa Fusarium root rot, it assumes paramount significance to unravel the distinct resistance mechanisms present in various alfalfa varieties. The objective of this study was to investigate mechanisms of disease resistance by examining the infection process, physiological and biochemical indices, and transcriptome sequencing of alfalfa inoculated with *F. acuminatum*.

## 2. Materials and Methods

### 2.1. Inoculation of Alfalfa Roots with Fusarium acuminatum

Alfalfa ‘Zhongmu No. 1, TMSS’, susceptible to root rot, and ‘Kangsai, TMSR’, resistant to root rot, were inoculated with *F. acuminatum* strain HM29-05, which was labeled with green fluorescent protein (GFP), using conidial suspension. Disinfected alfalfa seeds were placed with uniform spacing in a hydroponic incubator, covered, and incubated in an artificial growth chamber (MGC-400H, Shanghai Yiheng Instrument Co., Ltd., Shanghai, China). The cultivation conditions included temperature of 24 to 26 °C, light/dark cycle of 16 h/8 h, light intensity of 4200 lux, and 40% relative humidity. After 14 days of incubation, the roots were completely immersed in *F. acuminatum* conidial suspension at 1 × 10^7^ spore/mL for 30 min. Sterile water was used as a blank control. To investigate the infection process, alfalfa roots were sampled from three to five seedlings each time at 0.5, 1, 2, 3, 4, 5, 6, and 7 (dpi). The fluorescence of GFP-labeled *F. acuminatum* was measured under a laser confocal microscope (Leica, Weztlar, Germany) for one week.

The roots were frozen in liquid nitrogen and subsequently stored at −80 °C for physiological and biochemical measurements and transcriptome sequencing.

### 2.2. Determination of Physiological and Biochemical Indicators

Three biological replicates of root samples of ‘Kangsai’, designated asTMSR0, TMSR2, and TMSR3, and ‘Zhongmu No. 1’, designated asTMSS0, TMSS2, and TMSS3, were prepared at 0, 2, and 3 dpi. Measurement of soluble sugar, soluble protein, and malondialdehyde (MDA) contents, as well as superoxide dismutase (SOD) activity, were performed with a soluble sugar content kit, BCA assay kit for protein content, malondialdehyde content kit, and SOD-WST-8 method activity assay kit (Suzhou Gruise Biotechnology Co., Ltd., Suzhou, China). Soluble sugar contents were determined by anthrone colorimetry at 620 nm, soluble protein contents by the bicinchoninic acid (BCA) method at 562 nm, MDA contents by the thiobarbituric acid method, calculated as the difference in the 532 and 600 nm absorbances, and SOD activity was determined by the WST-8 method [33,34,35,36].

### 2.3. Transcriptome Sequencing

Root samples of ‘Kangsai’, designated as TMSR0, TMSR2, and TMSR3, and ‘Zhongmu No. 1’, designated as TMSS0, TMSS2, and TMSS3, were, respectively, collected at 0, 2, and 3 dpi and analyzed at Lianchuan Biotechnology Co., Ltd. (Hangzhou, China) for transcriptome sequencing. Each sample included three biological replicates, resulting in a total of 18 samples. Total RNA was isolated and purified using the TRIzol reagent (Invitrogen, Waltham, MA, USA) following the manufacturer’s instructions. The quantity and purity of each RNA sample were evaluated using a NanoDrop ND-1000 spectrophotometer (NanoDrop, Wilmington, DE, USA). The RNA integrity was assessed with a Bioanalyzer 2100 (Agilent, Santa Clara, CA, USA), with RIN number >7.0, and confirmed by electrophoresis on denaturing agarose gels, followed by library construction. The 2 × 150 bp paired-end sequencing (PE150) was performed on an Illumina Novaseq™ 6000 platform (LC-Bio Technology Co., Ltd., Hangzhou, China) following the vendor’s recommended protocol. Clean data were obtained by removing of adapters containing reads as well as low-quality raw reads. The Q30, GC content, and the level of sequence replication of the clean reads were calculated.

### 2.4. WGCNA Analysis

A gene co-expression network was constructed, encompassing a total of 21,833 genes using the R package WGCNA v1.0 (University of California, Los Angeles, CA, USA). Within this network, each gene module (module eigengene, ME) is characterized by a gene-specific attribute known as the first principal component for all genes nestled within a given module, and the expression of the ME is regarded as the representative of all genes in a given module. The MEs of the module can be used to calculate the correlation between characters on behalf of the module itself; the closer the correlation between a character and a module, the more related it is to the gene function of the module.

### 2.5. Quantitative Real-Time PCR Analysis

To verify the results of the transcriptome analysis, gene expression of the 18 samples and 10 differentially expressed genes (DEGs) were assessed using a real-time quantitative polymerase chain reaction (qPCR). Primers for qPCR (Appendix A) were designed at the NCBI (https://www.ncbi.nlm.nih.gov/, accessed on 22 May 2023) website, and β-actin was selected as the internal control [37]. RT-qPCR was performed on FQD-96A fluorescence quantitative PCR instrument (Hangzhou Bori LineGene9620, Hangzhou, China). The reaction conditions were as follows: 95 °C for 2 min, followed by 40 cycles of 95 °C for 15 s, 48 °C for 20 s, and 72 °C for 20 s. The target genes and internal parameters of each sample were assessed with four replicates per sample. Gene expression was determined using the 2^−ΔΔCT^ method. Gene expression was normalized against the control group, which was set to 1 for comparison between different groups.

### 2.6. Statistical Analysis

Data on the physiological and biochemical indices were analyzed using the Word Processing System (Kingsoft Software Co., Ltd., Beijing, China). The mean square error test was conducted using the Levene test in SPSS 27.0 statistical software (International Business Machines Corporation Co., Ltd., New York, NY, USA). *p* < 0.05 were considered statistically significant when assessing differences between two groups. GraphPad 9 (GraphPad Software Inc., San Diego, CA, USA) was used for drawing of graphs. Analysis of the transcriptome sequencing was performed mainly in OmicStudio (Lianchuan Biotechnology Co., Ltd., Hangzhou, China). Cytoscape v3.9.1(National Resource for Network Biology, Camden, NJ, USA) was used to visualize the gene interaction network.

## 3. Results

### 3.1. Fusarium acuminatum Infection on Medicago sativa

Laser confocal microscopy showed that most spores attached to the root surface germinated at day 1 after inoculation with *F. acuminatum* HM29-05, forming germ tubes extending toward various directions (Figure 1A). Hyphae were mainly apparent on hair roots, which later moved and cover the other root surface (Figure 1B). At 2 dpi, the hyphae started to penetrate into the epidermal cells of the roots. At the osmotic site, the hyphae formed an expanded structure like an appressorium, with some hyphae forming multiple such structures as they continued to grow and extend (Figure 1C). At 3 dpi, the hyphae expedited the growth, occupying large areas on the root surfaces, with most hyphae growing along the grooves of the epidermal cells and in the intercellular spaces (Figure 1D).

### 3.2. Physiological and Biochemical Activities after Plant Infection 

To assess the impact of *F. acuminatum* infection on alfalfa, root samples of alfalfa seedlings were collected at 0, 2, and 3 dpi and analyzed for physiological and biochemical activities using the indices related to root rot resistance. It was found that soluble sugar contents in alfalfa seedling roots significantly increased after infection by *F. acuminatum*. For the susceptible variety, soluble sugar contents of TMSS2 and TMSS3 increased by 29.23% and 87.39%, respectively, compared with TMSS0. For the resistant variety, soluble sugar contents in TMSR2 and TMSR3 were 14.35% and 29.60% higher than TMSR0, respectively. By comparing the two varieties, soluble sugar contents in ‘Kangsai’ roots were double those in Zhongmu No. 1 roots (Figure 2A, *p* < 0.05). There was a significant reduction in soluble protein contents. At 2 dpi, soluble protein content decreased significantly by 23.79% in ‘Zhongmu No. 1’ and 33.98% in ‘Kangsai’ (Figure 2B, *p* < 0.05). MDA contents in the roots first increased and then decreased, with the highest MDA levels in TMSS2 and TMSR2 at 2 dpi, which were 67.11% and 40.72% higher than those of TMSS0 and TMSR0, respectively. At 3 dpi, MDA contents of TMSS3 and TMSR3 decreased by 51.79 nmol∙g^−1^ FW and 43.41 nmol∙g^−1^ FW, respectively (Figure 2C, *p* < 0.05). SOD activity slightly increased in the three days, but not significantly. Compared with TMSS0, SOD activity in TMSS2 and TMSS3 increased by 21.38 U∙g^−1^ FW and 4.32 U∙g^−1^ FW, respectively, whereas, compared with TMSR0, SOD activity in TMSR2 and TMSR3 increased by 10.61 U∙g^−1^ FW and 19.04 U∙g^−1^ FW, respectively (Figure 2D, *p* < 0.05).

### 3.3. Transcriptome Analysis of TMSR and TMSS in Response to Fusarium acuminatum Infection

To explore the mechanism of root rot resistance in alfalfa, 18 root samples were collected at 0, 2, and 3 dpi from ‘Zhongmu No. 1’ (TMSS0, TMSS2, TMSS3) and ‘Kangsai’ (TMSR0, TMSR2, TMSR3) seedlings, which contained three biological replicates. RNA of the samples was sequenced, and 868,736,348 clean reads were obtained by high-throughput sequencing and filtering to remove low-quality reads, accounting for 93.82% of the total raw reads. The Q20 base percentage was over 99.98%, whereas the Q30 base percentage was more than 98.50%, and the GC content was between 42 and 45%. These data indicated high reliability and could be used for subsequent relevant analysis (Appendix A).

### 3.4. Identification of Genes with Changes in Expression following Fusarium acuminatum Infection in Resistant (TMSR) and Susceptible (TMSS) Alfalfa Varieties

There were significant changes in gene expression in TMSR and TMSS in response to *F. acuminatum* infection (Figure 3). A comparison of TMSR2 and TMSR0 showed that there were 1442 DEGs, including 634 up-regulated and 808 down-regulated genes. By comparing TMSR3 and TMSR0, 2650 significant DEGs were identified, including 1170 up-regulated and 1435 down-regulated genes. The TMSS2 vs. TMSS0 comparison showed 1359 significant DEGs were identified, of which 542 were up-regulated, and 817 were down-regulated. The comparison between TMSS3 and TMSS0 showed 2101 DEGs identified, including 1081 up-regulated and 1020 down-regulated genes. The results showed that there were more DEGs in both TMSR and TMSS at 3 dpi.

In addition, we observed that there were 1309, 1202, and 1296 DEGs between TMSR0 and TMSR0, TMSR2 and TMSR2, and TMSR3 and TMSR3, respectively, including 489, 490, and 379 up-regulated and 820, 712, and 917 down-regulated genes, respectively. Considering samples collected at the same time after F. acuminatum infection, the differences between the resistant and susceptible varieties were closely related to their resistance to Fusarium root rot.

### 3.5. Identification and Functional Annotation of DEGs in TMSR and TMSS in Response to Fusarium acuminatum Infection

We performed GO enrichment analysis on the above DEGs. The results showed that 5858 DEGs in TMSR were enriched in 3740 GO terms, involving 2016 biological processes, −463 cellular components, and 1261 molecular functions in the different GO categories (Figure 4A,B). Of these, the top five mostly significantly enriched GO terms in the biological process category were biological process (GO: 0008150, 632 DEGs), regulation of transcription, DNA-templated (GO: 0006355, 408 DEGs), transcription DNA-templated (GO: 0006351, 334 DEGs), oxidation-reduction process (GO: 0055114, 252 DEGs), and protein phosphorylation (GO: 0006468, 200 DEGs). In the cellular components category, the top five mostly enriched GO terms were nucleus (GO: 0005634, 1587 DEGs), cytoplasm (GO: 0005737, 932 DEGs), plasma membrane (GO: 0005886, 883 DEGs), integral component of membrane (GO: 0016021, 777 DEGs), and cytosol (GO: 0005829, 606 DEGs), whereas the most enriched GO terms in the molecular function category were protein binding (GO: 0005515, 670 DEGs), molecular function (GO: 0003674, 617 DEGs), ATP binding (GO: 0005524, 441 DEGs), DNA binding (GO: 0003677, 398 DEGs), and DNA-binding transcription factor activity (GO: 0003700, 332 DEGs).

A total of 4113 DEGs were enriched in 2945 GO terms in TMSS, involving 1635 in the biological process category, 363 in cell components, and 947 in molecular functions (Figure 4C,D). The top five GO terms were biological process (GO: 0008150, 378 DEGs), regulation of transcription, DNA-templated (GO: 0006355, 408 DEGs), transcription, DNA-templated (GO: 000631, 237 DEGs), defense response (GO: 0006952, 216 DEGs), and oxidation-reduction process (GO: 0055114, 166 DEGs). The top five cellular component terms were nucleus (GO: 0005634, 1081 DEGs), plasma membrane (GO: 0005886, 652 DEGs), cytoplasm (GO: 0005737, 652 DEGs), integral component of membrane (GO: 0016021, 515 DEGs), and cytosol (GO: 0005829, 606 DEGs), whereas the top five molecular functions were protein binding (GO: 0005515, 520 DEGs), molecular function (GO: 0003674, 408 DEGs), ATP binding (GO: 0005524, 311 DEGs), DNA binding (GO: 0003677, 301 DEGs), and DNA-binding transcription factor activity (GO: 0003700, 235 DEGs).

According to the above results, when *F. acuminatum* invades alfalfa, plants reacted and resisted the invasion through changing corresponding biological processes, transcriptional regulation, and redox processes. It is worth noting that defense reactions played a significant role in TMSS. The analysis of cell component enrichments showed that the resistance of alfalfa to *F. acuminatum* infection was mainly reflected at the cellular level. In addition, the DEGs were found to be involved in protein binding, molecular functions, DNA binding, and DNA-binding transcription factor activity, suggesting that transporters and related transcription factors played an important role in regulating the response of alfalfa to *Fusarium* spp.

Analysis of KEGG pathway enrichment showed that DEGs in both TMSS and TMSR were primarily involved in plant hormone signal transduction (ko: 04075), plant–pathogen interaction (ko: 04626), phenyl propionic biosynthesis (ko: 00940), and MAPK signaling pathway–plant (ko: 04016) pathways (Figure 4E–H).

### 3.6. Identification and Functional Annotation of Core Resistance Genes in TMSR and TMSS

To identify the core genes associated with disease resistance in alfalfa, we conducted WGCNA analysis of all DEGs in TMSR and TMSS. (Figure 5). The results showed that DEGs were present in 21 modules, and TMSR was used as the core to count them. The tan, dark-turquoise, green-yellow, turquoise, and cyan modules contained 82, 33, 99, 150, and 52 DEGs, respectively. The genes of the tan, dark-turquoise, and green-yellow modules were significantly down-regulated after infection with *F. acuminatum*. The expression of genes in the turquoise module showed a significant upward trend over time. However, the levels of gene expression in the cyan module first increased and then decreased, with the highest expression levels seen on the second day. These 416 genes were regarded as the core genes associated with root rot resistance in TMSR. To further clarify the relationships between these genes, we constructed their potential association network based on their correlations, which included genes belonging to the cytochrome P450 (MsG0580028496.01, MsG0580028497.01, MsG0880044995.01, MsG0780040711.01), MYB (MsG0280011264.01, MsG0780041774.01), ERF (MsG0080048712.01, MsG0280010711.01), NAC (MsG0480022079.01), and bZIP (MsG0180004956.01) enzyme and transcription factor families (Figure 6A–D).

### 3.7. Verification of DEGs

To verify the authenticity of the transcriptional data, 10 core transcription factors associated with root rot resistance in TMSR were randomly selected and verified using qPCR. The results showed that the expression of the 10 DEGs was consistent with the findings (FPKM values) of the transcriptome sequencing, thus confirming the accuracy of the transcriptome analysis (Figure 7).

## 4. Discussion

We have demonstrated that *F. acuminatum* completed the infection course within a span of three days following inoculation. On the first day, the pathogen achieved penetration into the epidermis of alfalfa. On the second day, distinctive swollen structures were observed. On the third day, mycelia grew rapidly in the intercellular spaces of the root epidermis. These findings were similar to *Verticillium dahliae* infection of potato, *Phytophthora sojae* infection of soybean, and *V. albo-atrum* infection of cotton *F. oxysporum* infection of muskmelon [38,39,40,41].

Soluble sugars represent the primary energy sources in plants, which also influence a plant’s resistance to diseases. In this study, we found that soluble sugar contents in both resistant and susceptible alfalfa varieties increased significantly following *F. acuminatum* infection, indicating plant responses to the infection. However, soluble sugar levels in resistant varieties were roughly twice as high as those in susceptible varieties. A previous study comparing root rot-resistant and root rot-susceptible varieties of sesame demonstrated that soluble sugar content of the resistant varieties (ranging from 3.22 to 8.38%) was higher than compared to susceptible varieties (1.18 to 3.97%) [42]. 

Intriguingly, this finding agrees with other studies on various diseases. For example, the content of soluble sugar in rice seedlings is positively correlated with blast resistance during the initial stage of fungal infection [43]. This association may be due to the invasion of pathogenic fungi inducing the hydrolysis of small-molecule polysaccharides in plants, leading to an increase in soluble sugar and activation of the pentose phosphate pathway. As metabolism accelerates, soluble sugar content declines. In addition, pathogen infection induces the expression of resistance-associated genes in the host, resulting in a sharp increase in sugar content. This surge in sugars provides additional metabolic substrates for the pentose phosphate pathway, fostering the increased production of secondary metabolites associated with disease resistance. These metabolites enhance plant defense and fortify resistance against pathogen invasion [44].

We have found that the expression patterns of five genes mirrored the trends observed in both resistant and susceptible varieties. This implies that these genes might be involved in alfalfa’s resistance to the root rot by participating in the production or transport of soluble sugar. That indicated that soluble sugar content is positively correlated with root rot resistance during the early stages of *F. acuminatum* infection. Therefore, elevated soluble sugar levels are an indicator of resistance against root rot in alfalfa.

Proteins constitute the fundamental building materials of the plant phenotype. During the early stage of pathogen infection in host plants, there is a significant alteration in soluble proteins present within infected plant cells and tissues. This was supported by the report that the levels of soluble proteins in infected plant tissues and cells tended to decrease. For instance, a study on the relationships between nutrient levels and resistance to black mole in potato showed a negative correlation between soluble protein content and disease resistance [45]. In maize, stronger resistance to rough dwarf disease was associated with greater reductions in soluble protein content [46]. This was consistent with our observation that soluble protein contents in both resistant and susceptible varieties decreased significantly when *F. acuminatum* invaded alfalfa. Notably, in susceptible variety, we observed more substantial down-regulation in gene expression, coupled with a more pronounced decline in soluble protein content. These dynamics may be attributed to shifts in tissue gene expression within host plant cells. During the early stage of infection, the expression of defense-related genes may be inhibited. Additionally, changes in protein ubiquitination could lead to protein degradation.

An elevated MDA content is a critical indicator of an increase in the peroxidation of membrane lipids. Previous studies showed that MDA levels in pumpkin leaves of powdery mildew-resistant varieties are higher than those in susceptible varieties [47]. Here, we have found that as time goes after *F. acuminatum* infection, MDA content in alfalfa increased progressively and decreased after reaching a peak on the second day, although it was always higher than that before infection. This is in agreement of the observation on root rot of radix astragali and ginseng [48,49]. 

MDA is a byproduct of membrane lipid peroxidation. Raised MDA levels can inflict damage on cell membranes, accelerate pathogen invasion, and ultimately compromise disease resistance. In the plant–pathogen interaction pathway (ko: 04626), the DEGs between TMSR2 vs. TMSR0 and TMSS2 vs. TMSS0 are generally up-regulated, actively participating in phenylpropionic biosynthesis (ko: 00940), which is known to be related to the plant stress response and disease resistance [50]. Subsequent studies showed that the expressions of MsG0280010711.01 and MsG0780040711.01 in resistant variety also reached a peak on the second day. Conversely, in the susceptible variety, the expression of these genes decreased significantly on the second day. These findings underscore the potential significance of this gene in conferring disease resistance in the ‘Kangsai’ variety.

Meanwhile, other defense mechanisms may be involved in alfalfa’s resistance, as observed in other disease systems. Notably, the cytochrome P450 family is usually involved in the metabolism of indole glucosinolate, which inhibits fungal diseases. For instance, localized increases in indole glucosinolate are consistently linked to increased expression of *CYP83B1* in *Brassica rapa.* infected by *Alternaria brassicicola* and *Botrytis cinerea*. *CYP83B1* is a key gene responsible for the biosynthesis and regulation of indole glucosinolate [51]. Similarly, we found that the transcription factors MsG0580028496.01 and MsG0580028497.01 as core transcription factors are associated with root rot resistance in resistant alfalfa variety. Additionally, the ethylene and salicylic acid metabolic pathways, in which MsG0280011264.01 and MsG0180004956.01 exhibited differential expressions between resistant and susceptible varieties, play important roles in plant disease resistance [52]. ERF transcription factors, for instance, promote resistance against fungal pathogens by activating the biosynthesis of indole glucosinolate [53].

Moreover, genes associated with disease resistance in alfalfa exert regulatory effects over the *F. acuminatum* invasion. We consider that MsG0780040711.01 (*CYP94A1*) may play a pivotal role in alfalfa’s resistance to root rot. In the resistant variety, we observed a substantial alteration in the expression of *CYP94A1*, with peak expression occurring at 2 dpi, whereas it remained unchanged in the susceptible variety changes. Several studies have indicated that during plant–pathogen interactions, *CYP94A1* can produce C-18 cuticle monomers in vitro, a process that might be integral to the mechanism of plant disease resistance [54]. 

We have also found that *F. acuminatum* infection activates many disease resistance pathways in alfalfa, including plant hormone signal transduction, plant–pathogen interaction, phenylpropionic biosynthesis, and MAPK signaling pathway in plants. These have been well established in many studies. For example, phenylalanine metabolism plays a pivotal role in stress resistance within plants. In the case of disease, pathogen infection leads to increased activity of phenylalanine ammonia lyases (PAL), which promote lignin synthesis in plants [55,56,57,58]. Our data revealed that *CYP94A1* might be involved in the phenylpropionic acid pathway, a novel finding in the context of our study.

## 5. Conclusions

Based on the results, we concluded that it takes around one day for *F. acuminatum* to penetrate alfalfa and establish in the epidermal cells at 2 dpi, which continues to expand its growth in the intercellular spaces thereafter. During this course, soluble sugar, soluble protein, and MDA contents exhibit significant changes after the colonization of *F. acuminatum* in both resistant and susceptible varieties of alfalfa. Soluble sugar content increases in both resistant and susceptible varieties over time, but the quantity is much higher in the resistant variety compared to the susceptible variety. Soluble protein contents significantly decrease, regardless of the susceptibility of alfalfa, whereas MDA contents first increased and then decreased, reaching them highest level at 2 dpi in resistant and susceptible varieties. The SOD activity is less affected by *F. acuminatum* infection. Gene expression is significantly changed between the resistant and susceptible plants at 3 dpi. Disease resistance is highly linked to pathways of plant hormone signal transduction, plant–pathogen interaction, phenylpropionic biosynthesis, and MAPK signaling. There are 10 proteins belonging to the cytochrome P450, MYB, ERF, NAC, and bZIP families that are closely associated with root rot resistance in TMSR. This work provides a new insight for studying the dynamics of infection court over time during infection and the physiological characteristics and transcriptome of alfalfa after *F. acuminatum* inoculation.

## Figures and Tables

**Figure 1 plants-12-03634-f001:**
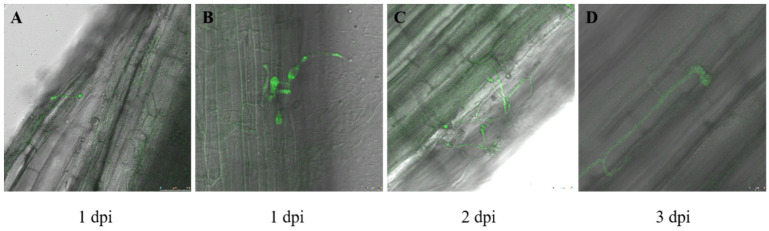
The initial infection process of alfalfa root system infected by *Fusarium acuminatum* HM29-05. At 1 day post inoculation (dpi), spores were attached to the root surface and germinated (**A**). At 1 dpi, the formation of germ tubes had grown in multiple directions (**B**). At 2 dpi, hyphae had invaded the epidermal cells of the roots and formed dilated structures similar to appressoria (**C**). At 3 dpi, the hyphae had grown rapidly within the intercellular spaces of the epidermis (**D**).

**Figure 2 plants-12-03634-f002:**
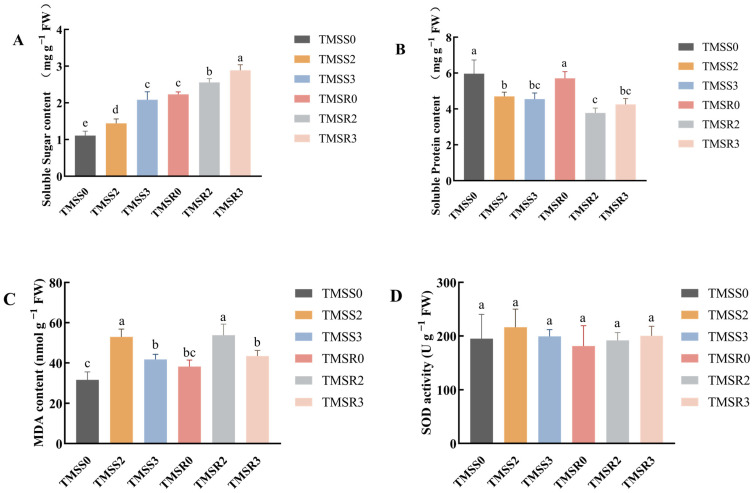
Physiological and biochemical indicators in the root system of alfalfa ‘Zhongmu No.1’ (TMSS, susceptible to root rot) and ‘Kangsai’ (TMSR, resistant to root rot) infected by *F. acuminatum* HM29-05. Sampling time points were designated as TMSS0 at 0 days post inoculation (dpi), TMSS2 at 2 dpi, and TMSS3 at 3 dpi for the TMSS group; TMSR0 at 0 dpi, TMSR2 at 2 dpi, and TMSR3 at 3 dpi for the TMSR group. Measurements included soluble sugar content (**A**), soluble protein content (**B**), MDA content (**C**), and SOD activity (**D**). Each value represents the average ± standard error of three replicates, with different letters indicating significant differences (*p* < 0.05).

**Figure 3 plants-12-03634-f003:**
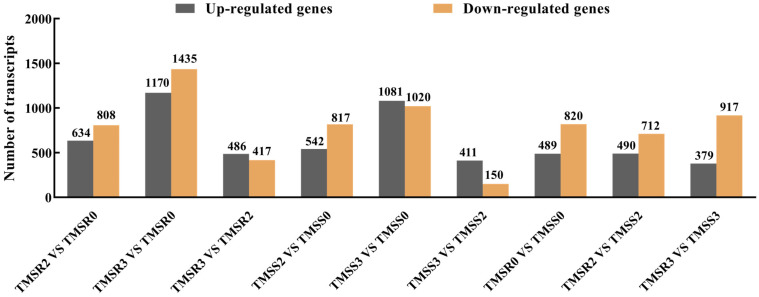
Differentially expressed transcripts in alfalfa seedling roots of varieties ‘Zhongmu No. 1’ (TMSS, susceptible to root rot) and ‘Kangsai’ (TMSR, resistant to root rot) infected by *Fusarium acuminatum* HM29-05. Sampling time points were designated as TMSS0 at 0 days post inoculation (dpi), TMSS2 at 2 dpi, and TMSS3 at 3 dpi for the TMSS group; TMSR0 at 0 dpi, TMSR2 at 2 dpi, and TMSR3 at 3 dpi for the TMSR group.

**Figure 4 plants-12-03634-f004:**
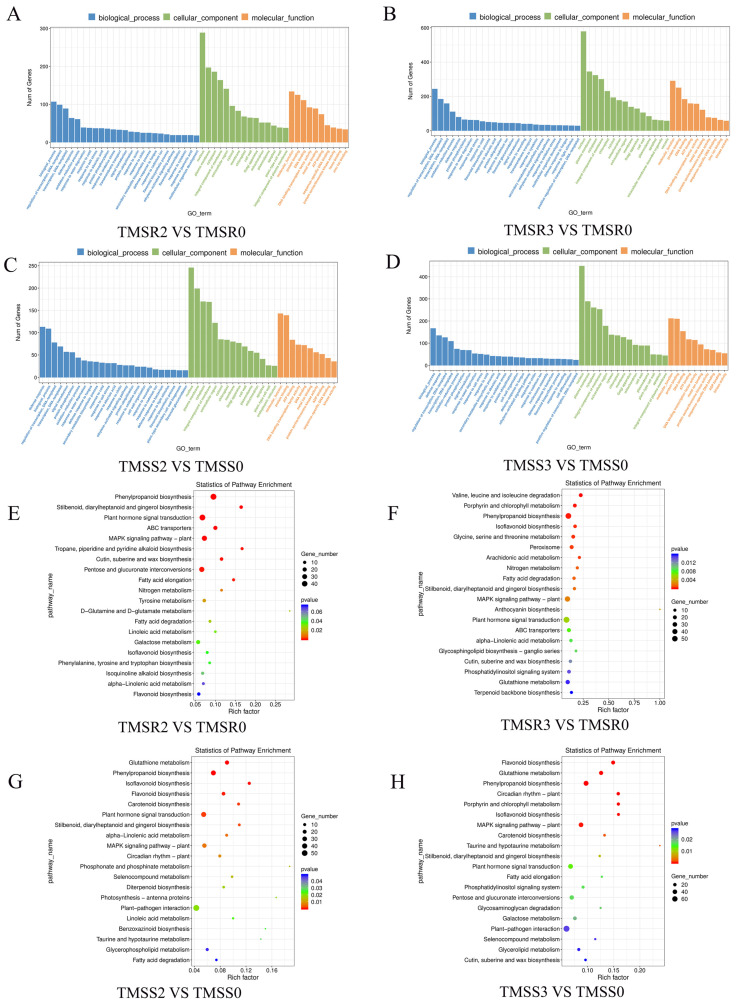
Gene ontology (GO) classification and Kyoto encyclopedia of genes and genomes (KEGG) enrichment analysis of deferential gene expressions (DEGs) between alfalfa varieties ‘Zhongmu No. 1’ (TMSS, the root rot-susceptible variety) and ‘Kangsai’ (TMSR, the root rot-resistant variety) inoculated with *Fusarium acuminatum*, sampled at 0, 2, and 3 days post inoculation (dpi), designated as TMSR0, TMSR2, and TMSR3 for the TMSR group, and TMSS0, TMSS2 and TMSS3 for the TMSS group. (**A**–**D**) GO classification of DEGs, and (**E**–**H**) KEGG enrichment analysis of DEGs.

**Figure 5 plants-12-03634-f005:**
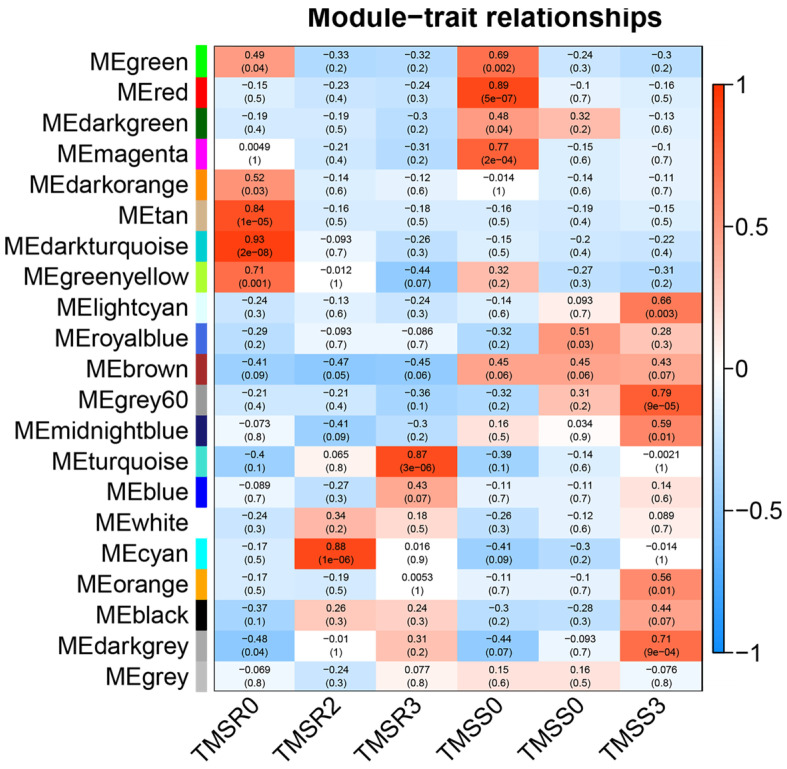
Weighted gene co-expression network analysis (WGCNA) of module eigengenes in corresponding modules in alfalfa seedling roots infected by *Fusarium acuminatum* HM29-05. Sampling time points included the susceptible variety ‘Zhongmu No. 1’ at 0 day post inoculation (dpi) (TMSS0), 2 dpi (TMSS2), and 3 dpi (TMSS3), and the resistant variety ‘Kangsai’ at 0 dpi (TMSR0), 2 dpi (TMSR2), and 3 dpi (TMSR3).

**Figure 6 plants-12-03634-f006:**
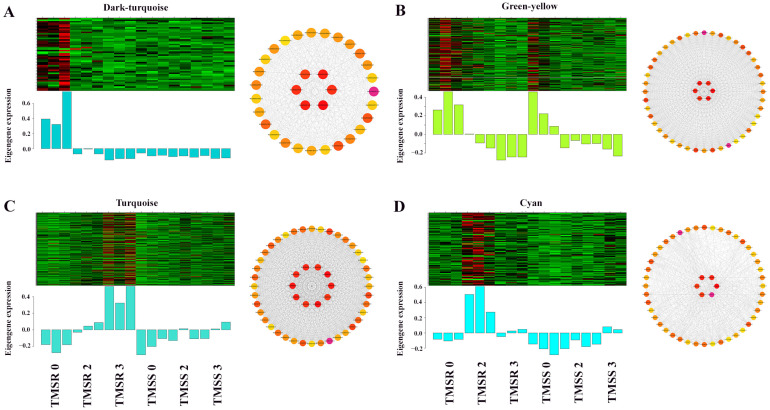
Weighted gene co-expression network analysis (WGCNA) of the key modules in alfalfa seedling roots infected by *Fusarium acuminatum* HM29-05. (**A**) Heatmap and bar plot of the expression of module eigengenes, with visualization of core genes of the dark-turquoise module. (**B**) Heatmap and bar plot of the expression of module eigengenes and visualization of the core genes of the green-yellow module. (**C**) Heatmap and bar plot of the expression of module eigengenes and visualization of the core genes of the turquoise module. (**D**) Heatmap and bar plot of the expression of module eigengenes and visualization of the core genes of the cyan module. Sampling time points included the susceptible variety ‘Zhongmu No. 1’ (TMSS) at 0 day post inoculation (dpi) (TMSS0), 2 dpi (TMSS2), and 3 dpi (TMSS3), and the resistant variety ‘Kangsai’ (TMSR) at 0 dpi (TMSR0), 2 dpi (TMSR2), and 3 dpi (TMSR3).

**Figure 7 plants-12-03634-f007:**
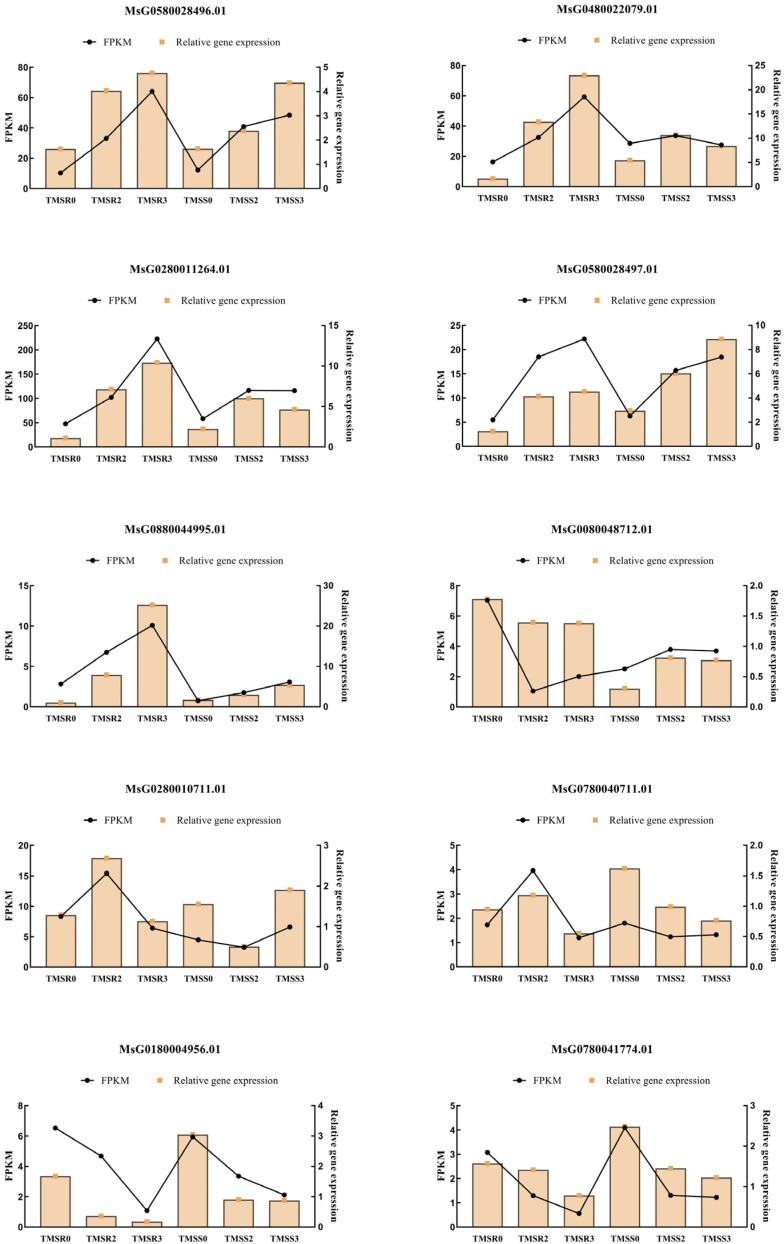
Validation of differentially expressed genes in alfalfa roots infected by *Fusarium acuminatum* HM29-05, conducted using quantitative polymerase chain reaction (qPCR). Sampling time points included the susceptible variety ‘Zhongmu No. 1’ at 0 day post inoculation (dpi) (TMSS0), 2 dpi (TMSS2), and 3 dpi (TMSS3), and the resistant variety ‘Kangsai’ at 0 dpi (TMSR0), 2 dpi (TMSR2), and 3 dpi (TMSR3).

## Data Availability

Data will be made available on request.

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
