# Peer review of "Identification of Crucial Genes and Regulatory Pathways in Alfalfa against Fusarium Root Rot"

_plants, 2023, doi:10.3390/plants12203634_

Round 1

Reviewer 1 Report

The manuscript Physiological, transcriptomic, and WGCNA analysis of key genes and regulatory pathways in alfalfa resistance to Fusarium root rot submitted by Wang et al. tried to investigate the mechanisms of alfalfa resistance to Fusarium root rot. They had labeled Fusarium acuminatum strain HM29-05 with GFP to track its infection process, and used transcriptomic and WGCNA analysis to show the interactions between plants and pathogens at the genetic level. However, before this manuscript become acceptable, it needs further deep revision.

1, Several of figures are quite unclearsuch as: figure 4 A - H, figure 5 B- E. The data and text in these images are not discernible.

2, In results 3.7. The authors had said that the qPCR of 15 DEGs was consistent with the findings, but they did not provide the results of the qPCR. We are also unable to ascertain which 15 DEGs they have chosen and the rationale behind their selection.

3, The Discussion section still requires a more in-depth analysis. The physiological and RNAseq data should deeply combine for analysis, deeply analysing the mechanisms of alfalfa resistance to Fusarium root rot.

No comments.

Author Response

For research article

Response to Reviewer 1 Comments

1. Summary

Thank you very much for your letter and the reviewers’ comments. Based on your comment and request, we have made extensive modification on the original manuscript. The point to point responds to reviewer’s comments are listed as following:

2. Questions for General Evaluation

Reviewer’s Evaluation

Response and Revisions

Does the introduction provide sufficient background and include all relevant references?

Can be improved

Are all the cited references relevant to the research?

Yes

Is the research design appropriate?

Can be improved

Are the methods adequately described?

Yes

Are the results clearly presented?

Can be improved

Are the conclusions supported by the results?

Can be improved

3. Point-by-point response to Comments and Suggestions for Authors

Comments 1:

Several of figures are quite unclear,such as: figure 4 A - H, figure 5 B- E. The data and text in these images are not discernible.

Response 1:

Thank you for pointing this out. I agree with this comment. Therefore, we have made new drawings and layouts of Figures 2, 3, 4, and 5 to achieve higher clarity.

Update location is as follows: Figure 2 on page 5; Figure 3 is on page 6; Figure 4. is on page 8; Figure 5 is on page 9.

Comments 2:

In results 3.7. The authors had said that the qPCR of 15 DEGs was consistent with the findings, but they did not provide the results of the qPCR. We are also unable to ascertain which 15 DEGs they have chosen and the rationale behind their selection.

Response 2:

Thank you for pointing this out. For the qPCR results, we have added Figure 6 in Chapter 3.7 (page 10). What we wanted to explain to you here is that because the layout of the picture of 15 genes will make the picture too small to affect the reading, we have chose the 10 core genes selected above for drawing.

Comments 3:

The Discussion section still requires a more in-depth analysis. The physiological and RNAseq data should deeply combine for analysis, deeply analysing the mechanisms of alfalfa resistance to Fusarium root rot.

Response 2:

Thank you for your suggestions. In chapter 4 (lines 372, 387 and 405), We have analyzed the physiological and RNAseq data obtained in the article and made some guesses. Since the genes we focused on are rarely reported on the molecular mechanism of plant disease resistance, there is no corresponding literature support.

4. Response to Comments on the Quality of English Language

Point 1: No comments.

Response 1:  Thanks.

5. Additional clarifications

Thank you for all your work.

Reviewer 2 Report

The study adresses the important question in the field of plant pathogen interaction. Authors should perforem PCA analysis  for the transcritopme data. 

Pleas check spelling erros

Author Response

For research article

Response to Reviewer 3 Comments

1. Summary

2. Questions for General Evaluation

Reviewer’s Evaluation

Response and Revisions

Does the introduction provide sufficient background and include all relevant references?

Yes

Are all the cited references relevant to the research?

Yes

Is the research design appropriate?

Yes

Are the methods adequately described?

Yes

Are the results clearly presented?

Can be improved

Are the conclusions supported by the results?

Can be improved

3. Point-by-point response to Comments and Suggestions for Authors

Comments 1:

Authors should modify the Figure 4, because the image resolution is too low. It's hard to read the text in the Figure.

Response 1:

Thank you for pointing this out. I agree with this comment. Therefore, I have made new drawings and layouts of Figures 2, 3, 4, and 5 to achieve higher clarity.

Update location is as follows: Figure 2 on page 6;Figure 3 is on page 7;Figure 4. is on page 8;Figure 5 is on page 11.

Comments 2:

  A more detailed description of Figure 3 legend should be done.

Response 2:

Thank you for pointing this out.I redrew Figure 3 and changed the legend "up" to "Up-regulated genes" and "down" to "Down-regulated genes".

Update location is as follows:Figure 3 is on page 7.

Comments 3:

The Discussion section still requires a more in-depth analysis. The physiological and RNAseq data should deeply combine for analysis, deeply analysing the mechanisms of alfalfa resistance to Fusarium root rot.

Response 2:

Thank you for your suggestions.In chapter 4 (lines 387, 405 and 426), I have analyzed the physiological and RNAseq data obtained in the article and made some guesses.Since the genes we focused on are rarely reported on the molecular mechanism of plant disease resistance, there is no corresponding literature support.

4. Response to Comments on the Quality of English Language

Point 1:No comments.

Response 1:  Thanks.

5. Additional clarifications

Thank you for all your work.

For review article

Response to Reviewer X Comments

1. Summary

Thank you very much for taking the time to review this manuscript. Please find the detailed responses below and the corresponding revisions/corrections highlighted/in track changes in the re-submitted files. [This is only a recommended summary. Please feel free to adjust it. We do suggest maintaining a neutral tone and thanking the reviewers for their contribution although the comments may be negative or off-target. If you disagree with the reviewer's comments please include any concerns you may have in the letter to the Academic Editor.]

2. Questions for General Evaluation

Reviewer’s Evaluation

Response and Revisions

Is the work a significant contribution to the field?

[Please give your response if necessary. Or you can also give your corresponding response in the point-by-point response letter. The same as below]

Is the work well organized and comprehensively described?

Is the work scientifically sound and not misleading?

Are there appropriate and adequate references to related and previous work?

Is the English used correct and readable?

3. Point-by-point response to Comments and Suggestions for Authors

Comments 1: [Paste the full reviewer comment here.]

Response 1: [Type your response here and mark your revisions in red] Thank you for pointing this out. I/We agree with this comment. Therefore, I/we have.[Explain what change you have made. Mention exactly where in the revised manuscript this change can be found – page number, paragraph, and line.]

“[updated text in the manuscript if necessary]”

Comments 2: [Paste the full reviewer comment here.]

Response 2: Agree. I/We have, accordingly, done/revised/changed/modified…..to emphasize this point. Discuss the changes made, providing the necessary explanation/clarification. Mention exactly where in the revised manuscript this change can be found – page number, paragraph, and line.]

“[updated text in the manuscript if necessary]”

4. Response to Comments on the Quality of English Language

Point 1:

Response 1:    (in red)

5. Additional clarifications

[Here, mention any other clarifications you would like to provide to the journal editor/reviewer.]

Reviewer 3 Report

The authors of the manuscript “Physiological, transcriptomic, and WGCNA analysis of key genes and regulatory pathways in alfalfa resistance to Fusarium root rot” intended to identify and perform the functional annotation of core resistance genes to Fusarium acuminatum infection in alfalfa.

The topic of this research is actual and may be interesting for readers.

The manuscript is generally well-written. However, there are some remarks that should be addressed.

1.     Authors should modify the Figure 4, because the image resolution is too low. It's hard to read the text in the Figure.

2.     A more detailed description of Figure 3 legend should be done.

Author Response

For research article

Response to Reviewer 3 Comments

1. Summary

Thank you very much for your letter and the reviewers’ comments. Based on your comment and request, we have made extensive modification on the original manuscript. The point to point responds to reviewer’s comments are listed as following:

2. Questions for General Evaluation

Reviewer’s Evaluation

Response and Revisions

Does the introduction provide sufficient background and include all relevant references?

Yes

Are all the cited references relevant to the research?

Yes

Is the research design appropriate?

Yes

Are the methods adequately described?

Yes

Are the results clearly presented?

Can be improved

Are the conclusions supported by the results?

Can be improved

3. Point-by-point response to Comments and Suggestions for Authors

Comments 1:

Authors should modify the Figure 4, because the image resolution is too low. It's hard to read the text in the Figure.

Response 1:

Thank you for pointing this out.We agree with this comment. Therefore, We have made new drawings and layouts of Figures 2, 3, 4, and 5 to achieve higher clarity.

Update location is as follows: Figure 2 on page 5; Figure 3 is on page 6; Figure 4. is on page 8; Figure 5 is on page 9.

Comments 2:

  A more detailed description of Figure 3 legend should be done.

Response 2:

Thank you for pointing this out.We have redrew Figure 3 and changed the legend "up" to "Up-regulated genes" and "down" to "Down-regulated genes".

Update location is as follows: Figure 3 is on page 6.

4. Response to Comments on the Quality of English Language

Point 1: No comments.

Response 1:  Thanks.

5. Additional clarifications

Thank you for all your work.
